# SEGEN: SAMPLE-ENSEMBLE GENETIC EVOLUTIONARY NETWORK MODEL

## ABSTRACT

Deep learning, a rebranding of deep neural network research works, has achieved a remarkable success in recent years. With multiple hidden layers, deep learning models aim at computing the hierarchical feature representations of the observational data. Meanwhile, due to its severe disadvantages in data consumption, computational resources, parameter tuning costs and the lack of result explainability, deep learning has also suffered from lots of criticism. In this paper, we will introduce a new representation learning model, namely "Sample-Ensemble Genetic Evolutionary Network" (SEGEN), which can serve as an alternative approach to deep learning models. Instead of building one single deep model, based on a set of sampled sub-instances, SEGEN adopts a genetic-evolutionary learning strategy to build a group of unit models generations by generations. The unit models incorporated in SEGEN can be either traditional machine learning models or the recent deep learning models with a much "narrower" and "shallower" architecture. The learning results of each instance at the final generation will be effectively combined from each unit model via diffusive propagation and ensemble learning strategies. From the computational perspective, SEGEN requires far less data, fewer computational resources and parameter tuning efforts, but has sound theoretic interpretability of the learning process and results. Extensive experiments have been done on several different real-world benchmark datasets, and the experimental results obtained by SEGEN have demonstrated its advantages over the state-of-the-art representation learning models.

## 1 INTRODUCTION

In recent years, deep learning, a rebranding of deep neural network research works, has achieved a remarkable success. The essence of deep learning is to compute the hierarchical feature representations of the observational data Goodfellow et al. (2016); LeCun et al. (2015). With multiple hidden layers, the deep learning models have the capacity to capture very good projections from the input data space to the objective output space, whose outstanding performance has been widely illustrated in various applications, including speech and audio processing Deng et al. (2013); Hinton et al. (2012), language modeling and processing Arisoy et al. (2012); Mnih & Hinton (2009), information retrieval Hill (2012); Salakhutdinov & Hinton (2009), objective recognition and computer vision LeCun et al. (2015), as well as multimodal and multi-task learning Weston et al. (2010; 2011). By this context so far, various kinds of deep learning models have been proposed already, including deep belief network Hinton et al. (2006), deep Boltzmann machine Salakhutdinov & Hinton (2009), deep neural network Jaeger (2002); Krizhevsky et al. (2012) and deep autoencoder model Vincent et al. (2010).

Meanwhile, deep learning models also suffer from several serious criticism due to their several severe disadvantages Zhou & Feng (2017a). Generally, learning and training deep learning models usually demands (1) a large amount of training data, (2) large and powerful computational facilities, (3) heavy parameter tuning costs, but lacks (4) theoretic explanation of the learning process and results. These disadvantages greatly hinder the application of deep learning models in many areas which cannot meet the requirements or requests a clear interpretability of the learning performance. Due to these reasons, by this context so far, deep learning research and application works are mostly carried out within/via the collaboration with several big technical companies, but the models proposed by them (involving hundreds of hidden layers, billions of parameters, and using a large cluster with thousands of server nodes Dean et al. (2012)) can hardly be applied in other real-world applications.

In this paper, we propose a brand new model, namely SEGEN (Sample-Ensemble Genetic Evolutionary Network), which can work as an alternative approach to the deep learning models. Instead of building one single model with a deep architecture, SEGEN adopts a genetic-evolutionary learning strategy to train a group of unit models generations by generations. Here, the unit models can be either traditional machine learning models or deep learning models with a much "narrower" and "shallower" structure. Each unit model will be trained with a batch of training instances sampled form the dataset. By selecting the good unit models from each generation (according to their performance on a validation set), SEGEN will evolve itself and create the next generation of unit modes with probabilistic genetic crossover and mutation, where the selection and crossover probabilities are highly dependent on their performance fitness evaluation. Finally, the learning results of the data instances will be effectively combined from each unit model via diffusive propagation and ensemble learning strategies. These terms and techniques mentioned here will be explained in great detail in Section 4. Compared with the existing deep learning models, SEGEN have several great advantages, and we will illustrate them from both the bionics perspective and the computational perspective as follows.

From the bionics perspective, SEGEN effectively models the evolution of creatures from generations to generations, where the creatures suitable for the environment will have a larger chance to survive and generate the offsprings. Meanwhile, the offsprings inheriting good genes from its parents will be likely to adapt to the environment as well. In the SEGEN model, each unit network model in generations can be treated as an independent creature, which will receive a different subsets of training instances and learn its own model variables. For the unit models suitable for the environment (i.e., achieving a good performance on a validation set), they will have a larger chance to generate their child models. The parent model achieving better performance will also have a greater chance to pass their variables to the child model.

From the computational perspective, SEGEN requires far less data and resources, and also has a sound theoretic explanation of the learning process and results. The unit models in each generation of SEGEN are of a much simpler architecture, learning of which can be accomplished with much less training data, less computational resources and less hyper-parameter tuning efforts. In addition, the training dataset pool, model hyper-parameters are shared by the unit models, and the increase of generation size (i.e., unit model number in each generation) or generation number (i.e., how many generation rounds will be needed) will not increase the learning resources consumption. The relatively "narrower" and "shallower" structure of unit models will also significantly enhance the interpretability of the unit models training process as well as the learning results, especially if the unit models are the traditional non-deep learning models. Furthermore, the sound theoretical foundations of genetic algorithm and ensemble learning will also help explain the information inheritance through generations and result ensemble in SEGEN.

In this paper, we will use network embedding problem Wang et al. (2016); Chang et al. (2015); Perozzi et al. (2014) (applying autoencoder as the unit model) as an example to illustrate the SEGEN model. Meanwhile, applications of SEGEN on other data categories (e.g., images and raw feature inputs) with CNN and MLP as the unit model will also be provided in Section 5.3. The following parts of this paper are organized as follows. The problem formulation is provided in Section 3. Model SEGEN will be introduced in Section 4, whose performance will be evaluated in Section 5. Finally, Section 2 introduces the related works and we conclude this paper in Section 6.

## 2   RELATED WORK

**Deep Learning Research and Applications**: The essence of deep learning is to compute hierarchical features or representations of the observational data Goodfellow et al. (2016); LeCun et al. (2015). With the surge of deep learning research and applications in recent years, lots of research works have appeared to apply the deep learning methods, like deep belief network Hinton et al. (2006), deep Boltzmann machine Salakhutdinov & Hinton (2009), Deep neural network Jaeger (2002); Krizhevsky et al. (2012) and Deep autoencoder model Vincent et al. (2010), in various applications, like speech and audio processing Deng et al. (2013); Hinton et al. (2012), language modeling and processing Arisoy et al. (2012); Mnih & Hinton (2009), information retrieval Hill (2012); Salakhutdinov & Hinton (2009), objective recognition and computer vision LeCun et al. (2015), as well as multimodal and multi-task learning Weston et al. (2010; 2011).

**Network Embedding**: Network embedding has become a very hot research problem recently, which can project a graph-structured data to the feature vector representations. In graphs, the relation can be treated as a translation of the entities, and many translation based embedding models have been proposed, like TransE Bordes et al. (2013), TransH Wang et al. (2014) and TransR Lin et al. (2015). In recent years, many network embedding works based on random walk model and deep learning models have been introduced, like Deepwalk Perozzi et al. (2014), LINE Tang et al. (2015), node2vec Grover & Leskovec (2016), HNE Chang et al. (2015) and DNE Wang et al. (2016). Perozzi et al. extends the word2vec model Mikolov et al. (2013) to the network scenario and introduce the Deepwalk algorithm Perozzi et al. (2014). Tang et al. Tang et al. (2015) propose to embed the networks with LINE algorithm, which can preserve both the local and global network structures. Grover et al. Grover & Leskovec (2016) introduce a flexible notion of a node's network neighborhood and design a biased random walk procedure to sample the neighbors. Chang et al. Chang et al. (2015) learn the embedding of networks involving text and image information. Chen et al. Chen & Sun (2016) introduce a task guided embedding model to learn the representations for the author identification problem.

## 3   PROBLEM FORMULATION

In this section, we will provide the definitions of several important terminologies, based on which we will define the network representation learning problem.

### 3.1   TERMINOLOGY DEFINITION

The SEGEN model will be illustrated based on the network representation learning problem in this paper, where the input is usually a large-sized network structured dataset.

**DEFINITION 1** (*Network Data*): *Formally, a network structured dataset can be represented as a graph $G = (\mathcal{V}, \mathcal{E})$, where $\mathcal{V}$ denotes the node set and $\mathcal{E}$ contains the set of links among the nodes.*

In the real-world applications, lots of data can be modeled as *networks*. For instance, online social media can be represented as a network involving users as the nodes and social connections as the links; e-commerce website can be denoted as a network with customer and products as the nodes, and purchase relation as the links; academic bibliographical data can be modeled as a network containing papers, authors as the nodes, and write/cite relationships as the links. Given a large-sized input network data $G = (\mathcal{V}, \mathcal{E})$, a group of sub-networks can be extracted from it, which can be formally represented as a sub-network set of $G$.

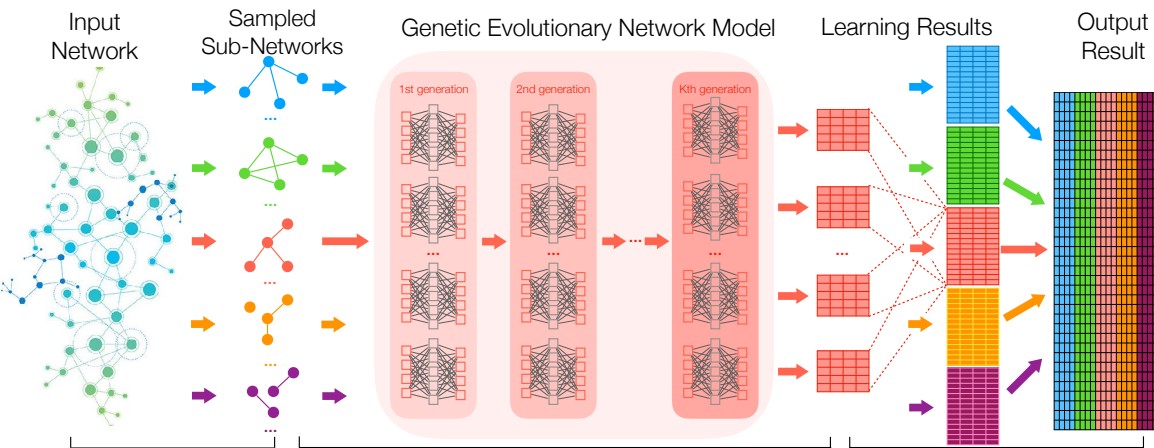

Figure 1: The SEGEN Framework.

**DEFINITION 2** *(Sub-network Set): Based on a certain sampling strategy, we can represent the set of sampled sub-networks from network $G$ as set $\mathcal{G} = \{g_1, g_2, \cdots, g_m\}$ of size $m$. Here, $g_i \in \mathcal{G}$ denotes a sub-network of $G$, and it can be represented as $g_i = (\mathcal{V}_{g_i}, \mathcal{E}_{g_i})$, where $\mathcal{V}_{g_i} \subseteq \mathcal{V}$, $\mathcal{E}_{g_i} \subseteq \mathcal{E}$ and $G \neq g_i$.*

In Section 4, we will introduce several different sampling strategies, which will be applied to obtained several different sub-network pools for unit model building and validation.

## 3.2 PROBLEM FORMULATION

**Problem Statement**: Based on the input network data $G = (\mathcal{V}, \mathcal{E})$, the *network representation learning* problem aims at learning a mapping $f : \mathcal{V} \to \mathbb{R}^d$ to project each node from the network to a low-dimensional feature space. There usually exist some requirements on mapping $f(\cdot)$, which should preserve the original network structure, i.e., closer nodes should have close representations; while disconnected nodes have different representations on the other hand.

## 4 PROPOSED METHODS

In this section, we will introduce the proposed framework SEGEN in detail. As shown in Figure 1, the proposed framework involves three steps: (1) network sampling, (2) sub-network representation learning, and (3) result ensemble. Given the large-scale input network data, framework SEGEN will sample a set of sub-networks, which will be used as the input to the *genetic evolutionary network* model for representation learning. Based on the learned results for the sub-networks, framework SEGEN will combine them together to obtain the final output result. In the following parts, we will introduce these three steps in great detail respectively.

### 4.1 NETWORK SAMPLING

In framework SEGEN, instead of handling the input large-scale network data directly, we propose to sample a subset (of set size $s$) of small-sized sub-networks (of a pre-specified sub-network size $k$) instead and learn the representation feature vectors of nodes based on the sub-networks. To ensure the learned representations can effectively represent the characteristics of nodes, we need to ensure the sampled sub-networks share similar properties as the original large-sized input network. As shown in Figure 1, 5 different types of network sampling strategies (indicated in 5 different colors) are adopted in this paper, and each strategy will lead to a group of small-sized sub-networks, which can capture both the local and global structures of the original network.

#### 4.1.1 BFS BASED NETWORK SAMPLING

Based on the input network $G = (\mathcal{V}, \mathcal{E})$, Breadth-First-Search (BFS) based network sampling strategy randomly picks a seed node from set $\mathcal{V}$ and performs BFS to expend to the unreached nodes. Formally, the neighbors of node $v \in \mathcal{V}$ can be denoted as set $\Gamma(v; 1) = \{u | u \in \mathcal{V} \wedge (u, v) \in \mathcal{E}\}$. After picking $v$, the sampling strategy will continue to randomly add $k - 1$ nodes from set $\Gamma(v; 1)$, if $|\Gamma(v; 1)| \geq k - 1$; otherwise, the sampling strategy will go to the 2-hop neighbors of $v$ (i.e., $\Gamma(v; 2) = \{u | \exists w \in \mathcal{V}, (u, w) \in \mathcal{E} \wedge (w, v) \in \mathcal{E} \wedge (u, v) \notin \mathcal{E}\}$) and so forth until the remaining $k - 1$ nodes are selected. In the case when the size of connected component that $v$ involves in is smaller than $k$, the strategy will further pick another seed node to do BFS from that node to finish the sampling of $k$ nodes. These sampled $k$ nodes together with the edges among them will form a sampled sub-network $g$, and all the $p$ sampled sub-networks will form the sub-network pool $\mathcal{G}^{\text{BFS}}$ (parameter $p$ denotes the pool size).

#### 4.1.2 DFS BASED NETWORK SAMPLING

Depth-First-Search (DFS) based network sampling strategy works in a very similar way as the BFS based strategy, but it adopts DFS to expand to the unreached nodes instead. Similar to the BFS method, in the case when the node connected component has size less than $k$, DFS sampling strategy will also continue to pick another node as the seed node to continue the sampling process. The sampled nodes together with the links among them will form the sub-networks to be involved in the final sampled sub-network pool $\mathcal{G}^{\text{DFS}}$ (of size $p$).

A remark to be added here: the sub-networks sampled via BFS can mainly capture the local network structure of nodes (i.e., the neighborhood), and in many of the cases they are star structured diagrams with the picked seed node at the center surrounded by its neighbors. Meanwhile, the sub-networks sampled with DFS are slightly different, which involve "deeper" network connection patterns. In the extreme case, the sub-networks sampled via DFS can be a path from the seed nodes to a node which is $(k-1)$-hop away.

### 4.1.3 HS BASED NETWORK SAMPLING

To balance between those extreme cases aforementioned, we introduce a Hybrid-Search (HS) based network sampling strategy by combining BFS and DFS. HS randomly picks seed nodes from the network, and reaches other nodes based on either BFS or DFS strategies with probabilities $p$ and $(1-p)$ respectively. For instance, in the sampling process, HS first picks node $v \in \mathcal{V}$ as the seed node, and samples a random node $u \in \Gamma(v; 1)$. To determine the next node to sample, HS will "toss a coin" with $p$ probability to sample nodes from $\Gamma(v; 1) \setminus \{u\}$ (i.e., BFS) and $1-p$ probability to sample nodes from $\Gamma(u; 1) \setminus \{v\}$ (i.e., DFS). Such a process continues until $k$ nodes are selected, and the sampled nodes together with the links among them will form the sub-network. We can represent all the sampled sub-networks by the HS based network sampling strategy as pool $\mathcal{G}^{\text{HS}}$.

These three network sampling strategies are mainly based on the connections among the nodes, and nodes in the sampled sub-networks are mostly connected. However, in the real-world networks, the connections among nodes are usually very sparse, and most of the node pairs are not connected. In the following part, we will introduce two other sampling strategies to handle such a case.

### 4.1.4 BIASED NODE SAMPLING

Instead of sampling sub-networks via the connections among them, the node sampling strategy picks the nodes at random from the network. Based on node sampling, the final sampled sub-network may not necessarily be connected and can involve many isolated nodes. Furthermore, uniform sampling of nodes will also deteriorate the network properties, since it treats all the nodes equally and fails to consider their differences. In this paper, we propose to adopt the *biased node* sampling strategy, where the nodes with more connections (i.e., larger degrees) will have larger probabilities to be sampled. Based on the connections among the nodes, we can represent the degree of node $v \in \mathcal{V}$ as $d(u) = |\Gamma(u; 1)|$, and the probabilities for $u$ to be sampled can be denoted as $p(u) = \frac{d(u)}{2|\mathcal{E}|}$. Instead of focusing on the local structures of the network, the sub-networks sampled with the *biased node sampling strategy* can capture more "global" structures of the input network. Formally, all the sub-networks sampled via this strategy can be represented as pool $\mathcal{G}^{\text{NS}}$.

### 4.1.5 BIASED EDGE SAMPLING

Another "global" sub-network sampling strategy is the edge based sampling strategy, which samples the edges instead of nodes. Here, uniform sampling of edges will be reduced to biased node selection, where high-degree nodes will have a larger probability to be involved in the sub-network. In this paper, we propose to adopt a *biased edge sampling strategy* instead. For each edge $(u, v) \in \mathcal{E}$, the probability for it to be sampled is actually proportional to $\frac{d(u)+d(v)}{2|\mathcal{E}|}$. The sampled edges together with the incident nodes will form a sub-network, and all the sampled sub-networks with *biased edge sampling strategy* can be denoted as pool $\mathcal{G}^{\text{ES}}$.

These two network sampling strategies can select the sub-structures of the input network from a global perspective, which can effectively capture the sparsity property of the input network. In the experiments to be introduced in Section 5, we will evaluate these different sampling strategies in detail.

## 4.2 GEN MODEL

In this part, we will focus on introducing the Genetic Evolutionary Network (GEN) model, which accepts each sub-network pool as the input and learns the representation feature vectors of nodes as the output. We will use $\mathcal{G}$ to represent the sampled pool set, which can be $\mathcal{G}^{\text{BFS}}$, $\mathcal{G}^{\text{DFS}}$, $\mathcal{G}^{\text{HS}}$, $\mathcal{G}^{\text{NS}}$ or $\mathcal{G}^{\text{ES}}$ respectively.

### 4.2.1 UNIT MODEL POPULATION INITIALIZATION

In the GEN model, there exist multiple generations of unit models, where the earlier generations will evolve and generate the later generations. Each generation will also involve a group of unit models, namely the unit model population. Formally, the initial generation of the unit models (i.e., the $1_{st}$ generation) can be represented as set $\mathcal{M}^1 = \{M_1^1, M_2^1, \cdots, M_m^1\}$ (of size $m$), where $M_i^1$ is a base unit model to be introduced in the following subsection. Formally, the variables involved in each unit model, e.g., $M_i^1$, can be denoted as vector $\theta_i^1$, which covers the weight and bias terms in the model (which will be treated as the model genes in the evolution to be introduced later). In the initialization step, the variables of each unit model are assigned with a random value generated from the standard normal distribution.

### 4.2.2 UNIT MODEL DESCRIPTION

In this paper, we will take network representation learning as an example, and propose to adopt the *correlated autoencoder* as the base model. We want to clarify again that the SEGEN framework is a general framework, and it works well for different types of data as well as different base models. For some other tasks or other learning settings, many other existing models, e.g., CNN and MLP to be introduced in Section 5.3, can be adopted as the base model as well.

Autoencoder is an unsupervised neural network model, which projects data instances from the original feature space to a lower-dimensional feature space via a series of non-linear mappings. Autoencoder model involves two steps: encoder

and decoder. The encoder part projects the original feature vectors to the objective feature space, while the decoder step recovers the latent feature representations to a reconstructed feature space.

Based on each sampled sub-network $g \in \mathcal{T}$, where $g = (\mathcal{V}_g, \mathcal{E}_g)$, we can represent the sub-network structure as an adjacency matrix $\mathbf{A}_g = \{0, 1\}^{|\mathcal{V}_g| \times |\mathcal{V}_g|}$, where $A_g(i, j) = 1$ iff $(v_i, v_j) \in \mathcal{E}_g$. Formally, for each node $v_i \in \mathcal{V}_g$, we can represent its raw feature as $\mathbf{x}_i = \mathbf{A}_g(i, :)$. Let $\mathbf{y}_i^1, \mathbf{y}_i^2, \cdots, \mathbf{y}_i^o$ be the corresponding latent feature representation of $\mathbf{x}_i$ at hidden layers $1, 2, \cdots, o$ in the encoder step. The encoding result in the objective feature space can be denoted as $\mathbf{z}_i \in \mathbb{R}^d$ of dimension $d$. In the decoder step, the input will be the latent feature vector $\mathbf{z}_i$, and the final output will be the reconstructed vector $\hat{\mathbf{x}}_i$ (of the same dimension as $\mathbf{x}_i$). The latent feature vectors at each hidden layers can be represented as $\hat{\mathbf{y}}_i^o, \hat{\mathbf{y}}_i^{o-1}, \cdots, \hat{\mathbf{y}}_i^1$. As shown in

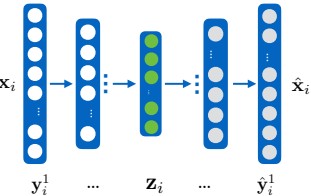

Figure 2: Autoencoder Model.

the architecture in Figure 2, the relationships among these variables can be represented with the following equations:

$$
\begin{cases}
\text{Encoder:} \\
\mathbf{y}_i^1 = \sigma(\mathbf{W}^1 \mathbf{x}_i + \mathbf{b}^1), \\
\mathbf{y}_i^k = \sigma(\mathbf{W}^k \mathbf{y}_i^{k-1} + \mathbf{b}^k), \forall k \in \{2, \cdots, o\}, \\
\mathbf{z}_i = \sigma(\mathbf{W}^{o+1} \mathbf{y}_i^o + \mathbf{b}^{o+1}).
\end{cases}
\quad
\begin{cases}
\text{Decoder:} \\
\hat{\mathbf{y}}_i^o = \sigma(\hat{\mathbf{W}}^{o+1} \mathbf{z}_i + \hat{\mathbf{b}}^{o+1}), \\
\hat{\mathbf{y}}_i^{k-1} = \sigma(\hat{\mathbf{W}}^k \hat{\mathbf{y}}_i^k + \hat{\mathbf{b}}^k), \forall k \in \{2, \cdots, o\}, \\
\hat{\mathbf{x}}_i = \sigma(\hat{\mathbf{W}}^1 \hat{\mathbf{y}}_i^1 + \hat{\mathbf{b}}^1).
\end{cases}
$$

The objective of traditional autoencoder model is to minimize the loss between the original feature vector $\mathbf{x}_i$ and the reconstructed feature vector $\hat{\mathbf{x}}_i$ of data instances. Meanwhile, for the network representation learning task, the learning task of nodes in the sub-networks are not independent but highly correlated. For the connected nodes, they should have closer representation feature vectors in the latent feature space; while for those which are isolated, their latent representation feature vectors should be far away instead. What's more, since the input feature vectors are extremely sparse (lots of the entries are 0s), simply feeding them to the model may lead to some trivial solutions, like $\mathbf{0}$ vector for both $\mathbf{z}_i$ and the decoded vector $\hat{\mathbf{x}}_i$. Therefore, we propose to extend the Autoencoder model to the correlated scenario for networks, and define the objective of the *correlated autoencoder* model as follows:

$$
\mathcal{L}_e(g) = \sum_{v_i \in \mathcal{V}_g} \left\| (\mathbf{x}_i - \hat{\mathbf{x}}_i) \odot \mathbf{b}_i \right\|_2^2 + \alpha \sum_{v_i, v_j \in \mathcal{V}_g, v_i \neq v_j} s_{i,j} \left\| \mathbf{z}_i - \mathbf{z}_j \right\|_2^2 + \beta \cdot \sum_{i=1}^{o} \left( \left\| \mathbf{W}^i \right\|_F^2 + \left\| \hat{\mathbf{W}}^i \right\|_F^2 \right),
$$

where $s_{i,j} = \begin{cases} +1, & \text{if } A_g(i, j) = 1, \\ -1, & \text{if } A_g(i, j) = 0. \end{cases}$ and $\alpha$, $\beta$ are the weights of the *correlation* and *regularization* terms respectively. Entries in weight vector $\mathbf{b}_i$ have value 1 except the entries corresponding to non-zero element in $\mathbf{x}_i$, which will be assigned with value $\gamma$ ($\gamma > 1$) to preserve these non-zero entries in the reconstructed vector $\hat{\mathbf{x}}_i$.

### 4.2.3 GENERATION MODEL LEARNING SETTING

Instead of fitting each unit model with all the sub-networks in the pool $\mathcal{G}$, in GEN, a set of sub-network training batches $\mathcal{T}_1, \mathcal{T}_2, \cdots, \mathcal{T}_m$ will be sampled for each unit model respectively in the learning process, where $|\mathcal{T}_i| = b, \forall i \in \{1, 2, \cdots, m\}$ are of the pre-defined batch size $b$. These batches may share common sub-networks as well, i.e., $\mathcal{T}_i \cap \mathcal{T}_j$ may not necessary be $\emptyset$. In the GEN model, the unit models learning process for each generation involves two steps: (1) generating the batches $\mathcal{T}_i$ from the pool set $\mathcal{G}$ for each unit model $M_i^1 \in \mathcal{M}^1$, and (2) learning the variables of the unit model $M_i^1$ based on sub-networks in batch $\mathcal{T}_i$. Considering that the unit models have a much smaller number of hidden layers, the learning time cost of each unit model will be much less than the deeper models on larger-sized networks. In Section 5, we will provide a more detailed analysis about the running time cost and space cost of SEGEN.

### 4.2.4 UNIT MODEL FITNESS EVALUATION AND SELECTION

The unit models in the generation set $\mathcal{M}^1$ can have different performance, due to (1) different initial variable values, and (2) different training batches in the learning process. In framework SEGEN, instead of applying "deep" models with multiple hidden layers, we propose to "deepen" the models in another way: "evolve the unit model into 'deeper' generations". A genetic algorithm style method is adopted here for evolving the unit models, in which the well-trained unit models will have a higher chance to survive and evolve to the next generation. To pick the well-trained unit models, we need to evaluate their performance, which is done with the validation set $\mathcal{V}$ sampled from the pool. For each unit model $M_k^1 \in \mathcal{M}^1$, based on the sub-networks in set $\mathcal{V}$, we can represent the introduced loss of the model as

$$
\mathcal{L}_c(M_k^1; \mathcal{V}) = \sum_{g \in \mathcal{V}} \sum_{v_i, v_j \in \mathcal{V}_g, v_i \neq v_j} s_{i,j} \left\| \mathbf{z}_{k,i}^1 - \mathbf{z}_{k,j}^1 \right\|_2^2,
$$

where $\mathbf{z}_{k,i}^1$ and $\mathbf{z}_{k,j}^1$ denote the learned latent representation feature vectors of nodes $v_i, v_j$ in the sampled sub-network $g$ and $s_{i,j}$ is defined based on $g$ in the same way as introduced before.

The probability for each unit model to be picked as the parent model for the *crossover* and *mutation* operations can be represented as

$$
p(M_k^1) = \frac{\exp^{-\mathcal{L}(M_k^1; \mathcal{V})}}{\sum_{M_i^1 \in \mathcal{M}^1} \exp^{-\mathcal{L}(M_i^1; \mathcal{V})}}.
$$

In the real-world applications, a normalization of the loss terms among these unit models is necessary. For the unit model introducing a smaller loss, it will have a larger chance to be selected as the parent unit model. Considering that the *crossover* is usually done based a pair of parent models, we can represent the pairs of parent models selected from set $\mathcal{M}^1$ as $\mathcal{P}^1 = \{(M_i^1, M_j^1)_k\}_{k \in \{1,2,\cdots,m\}}$, based on which we will be able to generate the next generation of unit models, i.e., $\mathcal{M}^2$.

### 4.2.5 UNIT MODEL CROSSOVER AND MUTATION

For the $k_{th}$ pair of parent unit model $(M_i^1, M_j^1)_k \in \mathcal{P}^1$, we can denote their genes as their variables $\theta_i^1, \theta_j^1$ respectively (since the differences among the unit models mainly lie in their variables), which are actually their chromosomes for crossover and mutation.

**Crossover**: In this paper, we propose to adopt the *uniform crossover* to get the chromosomes (i.e., the variables) of their child model. Considering that the parent models $M_i^1$ and $M_j^1$ can actually achieve different performance on the validation set $\mathcal{V}$, in the crossover, the unit model achieving better performance should have a larger chance to pass its chromosomes to the child model.

Formally, the chromosome inheritance probability for parent model $M_i^1$ can be represented as

$$p(M_i^1) = \frac{\exp^{-\mathcal{L}(M_i^1; \mathcal{V})}}{\exp^{-\mathcal{L}(M_i^1; \mathcal{V})} + \exp^{-\mathcal{L}(M_j^1; \mathcal{V})}}$$

Meanwhile, the chromosome inheritance probability for model $M_j^1$ can be denoted as $p(M_j^1) = 1 - p(M_i^1)$.

In the uniform crossover method, based on parent model pair $(M_i^1, M_j^1)_k \in \mathcal{P}^1$, we can represent the obtained child model chromosome vector as $\theta_k^2 \in \mathbb{R}^{|\theta^1|}$ (the superscript denotes the $2_{nd}$ generation and $|\theta^1|$ denotes the variable length), which is generated from the chromosome vectors $\theta_i^1$ and $\theta_j^1$ of the parent models. Meanwhile, the crossover choice at each position of the chromosomes vector can be represented as a vector $\mathbf{c} \in \{i, j\}^{|\theta^1|}$. The entries in vector $\mathbf{c}$ are randomly selected from values in $\{i, j\}$ with a probability $p(M_i^1)$ to pick value $i$ and a probability $p(M_j^1)$ to pick value $j$ respectively. The $l_{th}$ entry of vector $\theta_k^2$ before mutation can be represented as

$$\hat{\theta}_k^2(l) = \mathbb{1}\left(c(l) = i\right) \cdot \theta_i^1(l) + \mathbb{1}\left(c(l) = j\right) \cdot \theta_j^1(l),$$

where indicator function $\mathbb{1}(\cdot)$ returns value 1 if the condition is True; otherwise, it returns value 0.

**Mutation**: The variables in the chromosome vector $\hat{\theta}_k^2(l) \in \mathbb{R}^{|\theta^1|}$ are all real values, and some of them can be altered, which is also called *mutation* in traditional genetic algorithm. Mutation happens rarely, and the chromosome mutation probability is $\gamma$ in the GEN model. Formally, we can represent the mutation indicator vector as $\mathbf{m} \in \{0, 1\}^d$, and the $l_{th}$ entry of vector $\theta_k^2$ after mutation can be represented as

$$\theta_k^2(l) = \mathbb{1}\left(m(l) = 0\right) \cdot \hat{\theta}_k^2(l) + \mathbb{1}\left(c(l) = 1\right) \cdot rand(0, 1),$$

where $rand(0, 1)$ denotes a random value selected from range $[0, 1]$. Formally, the chromosome vector $\theta_k^2$ defines a new unit model with knowledge inherited form the parent models, which can be denoted as $M_k^2$. Based on the parent model set $\mathcal{P}^1$, we can represent all the newly generated models as $\mathcal{M}^2 = \{M_k^2\}_{(M_i^1, M_j^1)_k \in \mathcal{P}^1}$, which will form the $2_{nd}$ generation of unit models.

### 4.3 RESULT ENSEMBLE

Based on the models introduced in the previous subsection, in this part, we will introduce the hierarchical result ensemble method, which involves two steps: (1) *local ensemble* of results for the sub-networks on each sampling strategies, and (2) *global ensemble* of results obtained across different sampling strategies.

### 4.3.1 LOCAL ENSEMBLE

Based on the sub-network pool $\mathcal{G}$ obtained via the sampling strategies introduced before, we have learned the $K_{th}$ generation of the GEN model $\mathcal{M}^K$ (or $\mathcal{M}$ for simplicity), which contains $m$ unit models. In this part, we will introduce how to fuse the learned representations from each sub-networks with the unit models. Formally, given a sub-network $g \in \mathcal{G}$ with node set $\mathcal{V}_g$, by applying unit model $M_j \in \mathcal{M}$ to $g$, we can represent the learned representation for node $v_q \in \mathcal{V}_g$ as vector $\mathbf{z}_{j,q}$, where $q$ denotes the unique node index in the original complete network $G$ before sampling. For the nodes $v_p \notin \mathcal{V}_g$, we can denote its representation vector $\mathbf{z}_{j,p} = \mathbf{null}$, which denotes a dummy vector of length $d$. Formally, we will be able represent the learned representation feature vector for node $v_q$ as

$$\mathbf{z}_q = \bigsqcup_{g \in \mathcal{G}, M_j \in \mathcal{M},} \mathbf{z}_{j,q}, \tag{1}$$

where operator $\sqcup$ denotes the concatenation operation of feature vectors.

Considering that in the network sampling step, not all nodes will be selected in sub-networks. For the nodes $v_p \notin \mathcal{V}_g, \forall g \in \mathcal{G}$, we will not be able to learn its representation feature vector (or its representation will be filled with a list of dummy

empty vector). Formally, we can represent these non-appearing nodes as set $\mathcal{V}_n = \mathcal{V} \setminus \bigcup_{g \in \mathcal{G}} \mathcal{V}_g$. In this paper, to compute the representation for these nodes, we propose to propagate the learned representation from their neighborhoods to them instead. Formally, given node $v_p \in \mathcal{V}_n$ and its neighbor set $\Gamma(v_p) = \{v_o | v_o \in \mathcal{V} \wedge (u, v_p) \in \mathcal{E}\}$, if there exists node in $\Gamma(v_p)$ with non-empty representation feature vector, we can represent the propagated representation for $v_p$ as

$$\mathbf{z}_p = \frac{1}{N} \sum_{v_o \in \Gamma(v_p)} \mathbb{1}(v_o \notin \mathcal{V}_n) \cdot \mathbf{z}_o, \tag{2}$$

where $N = \sum_{v_o \in \Gamma(v_p)} \mathbb{1}(v_o \notin \mathcal{V}_n)$. In the case that $\Gamma(v_p) \subset \mathcal{V}_n$, random padding will be applied to get the representation vector $\mathbf{z}_p$ for node $v_p$.

### 4.3.2 GLOBAL ENSEMBLE

Generally, these different network sampling strategies introduced at the beginning in Section 4.1 captures different local/global structures of the network, which will all be useful for the node representation learning. In the global result ensemble step, we propose to group these features together as the output.

Formally, based on the *BFS*, *DFS*, *HS*, *biased node* and *biased edge* sampling strategies, to differentiate their learned representations for nodes (e.g., $v_q \in \mathcal{V}$), we can denoted their representation feature vectors as $\mathbf{z}_q^{\text{BFS}}$, $\mathbf{z}_q^{\text{DFS}}$, $\mathbf{z}_q^{\text{HS}}$, $\mathbf{z}_q^{\text{NS}}$ and $\mathbf{z}_q^{\text{ES}}$ respectively. In the case that node $v_q$ has never appeared in any sub-networks in any of the sampling strategies, its corresponding feature vector can be denoted as a dummy vector filled with 0s. In the global ensemble step, we propose to linearly sum the feature vectors to get the fuses representation $\bar{\mathbf{z}}_q$ as follows:

$$\bar{\mathbf{z}}_q = \sum_{i \in \{\text{BFS,DFS,HS,NS,ES}\}} w^i \cdot \mathbf{z}_q^i.$$

Learning of the weight parameters $w^{\text{BFS}}$, $w^{\text{DFS}}$, $w^{\text{HS}}$, $w^{\text{NS}}$ and $w^{\text{ES}}$ is feasible with the complete network structure, but it may introduce extra time costs and greatly degrade the efficiency SEGEN. In this paper, we will simply assign them with equal value, i.e., $\bar{\mathbf{z}}_q$ is an average of $\mathbf{z}_q^{\text{BFS}}$, $\mathbf{z}_q^{\text{DFS}}$, $\mathbf{z}_q^{\text{HS}}$, $\mathbf{z}_q^{\text{NS}}$ and $\mathbf{z}_q^{\text{ES}}$ learned with different sampling strategies.

## 4.4 MODEL ANALYSIS

In this section, we will analyze the proposed model SEGEN regarding its performance, running time and space cost, which will also illustrate the advantages of SEGEN compared with the other existing deep learning models.

### 4.4.1 PERFORMANCE ANALYSIS

Model SEGEN, in a certain sense, can also be called a "deep" model. Instead of stacking multiple hidden layers inside one single model like existing deep learning models, SEGEN is deep since the unit models in the successive generations are generated by a namely "evolutionary layer" which performs the *validation*, *selection*, *crossover*, and *mutation* operations connecting these generations. Between the generations, these "evolutionary operations" mainly work on the unit model variables, which allows the immigration of learned knowledge from generation to generation. In addition, via these generations, the last generation in SEGEN can also capture the overall patterns of the dataset. Since the unit models in different generations are built with different sampled training batches, as more generations are involved, the dataset will be samples thoroughly for learning SEGEN. There have been lots of research works done on analyzing the convergence, performance bounds of genetic algorithms Rudolph (1994), which can provide the theoretic foundations for SEGEN.

Due to the difference in parent model selection, crossover, mutation operations and different sampled training batches, the unit models in the generations of SEGEN may perform quite differently. In the last step, SEGEN will effectively combine the learning results from the multiple unit models together. With the diverse results combined from these different learning models, SEGEN is able to achieve better performance than each of the unit models, which have been effectively demonstrated in Zhou et al. (2002).

### 4.4.2 SPACE AND TIME COMPLEXITY ANALYSIS

According the the model descriptions provided in Section 4, we summarize the key parameters used in SEGEN as follows, which will help analyze its space and time complexity.

- *Sampling*: Original data size: $n$. Sub-instance size: $n'$. Pool size: $p$.
- *Learning*: Generation number: $K$. Population size: $m$. Feature vector size: $d$. Training/Validation batch size: $b$.

Here, we will use network structured data as an example to analyze the space and time complexity of the SEGEN model.

**Space Complexity**: Given a large-scale network with $n$ nodes, the space cost required for storing the whole network in a matrix representation is $O(n^2)$. Meanwhile, via network sampling, we can obtain a pool of sub-networks, and the space required for storing these sub-networks takes $O\left(p(n')^2\right)$. Generally, in application of SEGEN, $n'$ can take very small number, e.g., 50, and $p$ can take value $p = c \cdot \frac{n}{n'}$ ($c$ is a constant) so as to cover all the nodes in the network. In such a case, the space cost of SEGEN will be linear to $n$, $O(cn'n)$, which is much smaller than $O(n^2)$.

**Time Complexity**: Depending on the specific unit models used in composing SEGEN, we can represent the introduced time complexity of learn one unit model with the original network with $n$ nodes as $O(f(n))$, where $f(n)$ is usually a high-order function. Meanwhile, for learning SEGEN on the sampled sub-networks with $n'$ nodes, all the introduced time cost

Table 1: Representation Learning Experiment Results Comparison on Foursquare Network Dataset.

| Network Recovery | AUC | | | Prec@500 | | | Community Detection | Density | | | Silhouette | | |
|---|---|---|---|---|---|---|---|---|---|---|---|---|---|
| | 1 | 5 | 10 | 1 | 5 | 10 | | 5 | 25 | 50 | 5 | 25 | 50 |
| SEGEN(PS2) | **0.909**(2) | **0.909**(2) | **0.909**(2) | **0.872**(2) | **0.642**(3) | **0.530**(3) | SEGEN(PS3) | **0.875**(2) | **0.550**(2) | **0.792**(3) | **0.353**(2) | **0.206**(2) | **0.208**(3) |
| SEGEN(PS1) | 0.817(6) | 0.819(6) | 0.818(6) | 0.772(5) | **0.400**(4) | **0.266**(4) | SEGEN(PS1) | 0.792(6) | **0.477**(4) | **0.742**(4) | **0.317**(4) | 0.188(3) | **0.156**(5) |
| SEGEN-HS(PS2) | **0.935**(1) | **0.936**(1) | **0.936**(1) | **0.852**(4) | **0.388**(5) | 0.000(-) | SEGEN-HS(PS3) | **0.812**(5) | 0.385(11) | **0.705**(5) | 0.252(10) | 0.056(6) | **0.166**(4) |
| SEGEN-BFS(PS2) | **0.860**(4) | **0.859**(4) | **0.858**(4) | 0.428(10) | 0.000(-) | 0.000(-) | SEGEN-BFS(PS3) | 0.746(7) | 0.425(8) | 0.587(6) | 0.206(11) | 0.022(10) | 0.108(6) |
| SEGEN-DFS(PS2) | **0.881**(3) | **0.882**(3) | **0.881**(3) | **0.965**(1) | **0.814**(2) | **0.648**(2) | SEGEN-DFS(PS3) | **0.860**(4) | **0.532**(3) | 0.436(11) | **0.280**(9) | 0.017(11) | -0.006(11) |
| SEGEN-NS(PS2) | 0.801(7) | 0.797(7) | 0.797(7) | 0.256(11) | 0.002(10) | 0.002(9) | SEGEN-NS(PS3) | **0.871**(3) | **0.425**(8) | **0.824**(2) | **0.327**(3) | **0.060**(5) | **0.294**(2) |
| SEGEN-ES(PS2) | **0.820**(5) | **0.822**(5) | **0.822**(5) | **0.872**(2) | **0.872**(1) | **0.872**(1) | SEGEN-ES(PS3) | **0.948**(1) | **0.933**(1) | **0.924**(1) | **0.482**(1) | **0.429**(1) | **0.407**(1) |
| LINE | 0.536(9) | 0.537(9) | 0.537(9) | 0.712(6) | 0.268(9) | 0.172(7) | LINE | 0.695(8) | 0.443(6) | 0.478(8) | **0.311**(5) | 0.046(8) | 0.082(8) |
| DEEPWALK | 0.536(9) | 0.537(9) | 0.537(9) | 0.686(9) | 0.308(7) | 0.184(6) | DEEPWALK | 0.695(8) | **0.449**(5) | 0.485(7) | **0.311**(5) | 0.042(9) | 0.082(8) |
| NODE2VEC | 0.538(8) | 0.540(8) | 0.539(8) | 0.692(8) | 0.299(8) | 0.162(8) | NODE2VEC | 0.691(11) | 0.419(10) | 0.469(9) | 0.297(8) | **0.066**(4) | 0.070(10) |
| HPE | 0.536(9) | 0.537(9) | 0.537(9) | 0.708(7) | 0.354(6) | **0.188**(5) | HPE | 0.695(8) | 0.431(7) | 0.465(10) | **0.311**(5) | 0.051(7) | 0.089(7) |

will be $O\left(Km(b \cdot f(n') + d \cdot n')\right)$, where term $d \cdot n'$ (an approximation to variable size) represents the cost introduced in the unit model crossover and mutation about the model variables. Here, by assigning $b$ with a fixed value $b = c \cdot \frac{n}{n'}$, the time complexity of SEGEN will be reduced to $O\left(Kmc\frac{f(n')}{n'} \cdot n + Kmdn'\right)$, which is linear to $n$.

### 4.4.3 ADVANTAGES OVER DEEP LEARNING MODELS

Compared with existing deep learning models based on the whole dataset, the advantages of SEGEN are summarized below:

- *Less Data for Unit Model Learning*: For each unit model, which are of a "shallow" and "narrow" structure (shallow: less or even no hidden layers, narrow: based on sampled sub-instances with a much smaller size), which needs far less variables and less data for learning each unit model.
- *Less Computational Resources*: Each unit model is of a much simpler structure, learning process of which consumes far less computational resources in both time and space costs.
- *Less Parameter Tuning*: SEGEN can accept both deep (in a simpler version) and shallow learning models as the unit model, and the hyper-parameters can also be shared among the unit models, which will lead to far less hyper-parameters to tune in the learning process.
- *Sound Theoretic Explanation*: The unit learning model, genetic algorithm and ensemble learning (aforementioned) can all provide the theoretic foundation for SEGEN, which will lead to sound theoretic explanation of both the learning result and the SEGEN model itself.

## 5 EXPERIMENTS

To test the effectiveness of the proposed model, extensive experiments will be done on several real-world network structured datasets, including social networks, images and raw feature representation datasets. In this section, we will first introduce the detailed experimental settings, covering experimental setups, comparison methods, evaluation tasks and metrics for the social network representation learning task. After that, we will show its convergence analysis, parameter analysis and the main experimental results of SEGEN on the social network datasets. Finally, we will provide the experiments SEGEN based on the image and raw feature representation datasets involving CNN and MLP as the unit models respectively.

### 5.1 SOCIAL NETWORK DATASET EXPERIMENTAL SETTINGS

#### 5.1.1 EXPERIMENTAL SETUP

The network datasets used in the experiments are crawled from two different online social networks, Twitter and Foursquare, respectively. The Twitter network dataset involves $5,120$ users and $130,576$ social connections among the user nodes. Meanwhile, the Foursquare network dataset contains $5,392$ users together with the $55,926$ social links connecting them. According to the descriptions of SEGEN, based on the complete input network datasets, a set of sub-networks are randomly sampled with network sampling strategies introduced in this paper, where the sub-network size is denoted as $n'$, and the pool size is controlled by $p$. Based on the training/validation batches sampled sub-network pool, $K$ generations of unit models will be built in SEGEN, where each generation involves $m$ unit models (convergence analysis regarding parameter $K$ is available in Section 7.1.1). Finally, the learning results at the ending generation will be effectively combined to generate the ensemble output. For the nodes which have never been sampled in any sub-networks, their representations can be learned with the diffusive propagation from their neighbor nodes introduced in this paper. The learned results by SEGEN will be evaluated with two application tasks, i.e., network recovery and community detection respectively. The detailed parameters sensitivity analysis is also available in Section 7.1.2.

#### 5.1.2 COMPARISON METHODS

The network representation learning comparison models used in this paper are listed as follows

Table 2: Representation Learning Experiment Results Comparison on Twitter Network Dataset.

| Network Recovery | AUC | | | Prec@500 | | | Community Detection | Density | | | Silhouette | | |
|---|---|---|---|---|---|---|---|---|---|---|---|---|---|
| | 1 | 5 | 10 | 1 | 5 | 10 | | 5 | 25 | 50 | 5 | 25 | 50 |
| SEGEN(PS4) | **0.879**(1) | **0.881**(1) | **0.881**(1) | **0.914**(3) | **0.638**(3) | **0.370**(3) | SEGEN(PS5) | **0.980**(2) | **0.845**(3) | **0.770**(3) | **0.566**(4) | **0.353**(3) | **0.341**(2) |
| SEGEN(PS1) | **0.814**(4) | **0.813**(4) | **0.814**(4) | **0.606**(4) | **0.194**(4) | **0.102**(4) | SEGEN(PS1) | 0.786(7) | **0.751**(4) | **0.753**(4) | 0.481(10) | **0.328**(4) | **0.318**(4) |
| SEGEN-HS(PS4) | **0.862**(2) | **0.863**(2) | **0.863**(2) | **0.594**(5) | 0.000(-) | 0.000(-) | SEGEN-HS(PS5) | 0.967(6) | 0.171(11) | 0.116(11) | **0.549**(5) | -0.021(11) | -0.032(10) |
| SEGEN-BFS(PS4) | **0.846**(3) | **0.846**(3) | **0.846**(3) | 0.570(6) | 0.000(-) | 0.000(-) | SEGEN-BFS(PS5) | **0.973**(4) | **0.973**(2) | **0.886**(2) | **0.622**(2) | **0.517**(2) | **0.330**(3) |
| SEGEN-DFS(PS4) | 0.503(10) | 0.508(10) | 0.507(10) | 0.268(10) | 0.000(-) | 0.000(-) | SEGEN-DFS(PS5) | **0.994**(1) | 0.338(9) | 0.143(10) | 0.532(7) | 0.237(6) | -0.023(9) |
| SEGEN-NS(PS4) | **0.794**(5) | **0.794**(5) | **0.795**(5) | **0.962**(2) | **0.824**(2) | **0.688**(2) | SEGEN-NS(PS5) | **0.979**(3) | **0.981**(1) | **0.965**(1) | **0.597**(3) | **0.525**(1) | **0.461**(1) |
| SEGEN-ES(PS4) | 0.758(6) | 0.760(6) | 0.760(6) | **1.000**(1) | **1.000**(1) | **1.000**(1) | SEGEN-ES(PS5) | **0.970**(5) | 0.525(8) | 0.482(8) | **0.693**(1) | **0.284**(5) | -0.059(11) |
| LINE | 0.254(11) | 0.254(11) | 0.253(11) | 0.106(11) | 0.018(7) | 0.006(7) | LINE | 0.524(11) | 0.324(10) | 0.251(9) | 0.465(11) | -0.012(9) | -0.012(7) |
| DEEPWALK | 0.533(9) | 0.531(9) | 0.532(9) | 0.524(9) | 0.146(6) | 0.070(6) | DEEPWALK | 0.545(10) | 0.542(7) | 0.503(7) | 0.492(9) | 0.173(8) | 0.150(6) |
| NODE2VEC | 0.704(7) | 0.703(7) | 0.704(7) | 0.528(8) | 0.012(8) | 0.000(-) | NODE2VEC | 0.697(8) | **0.693**(5) | **0.694**(5) | 0.530(8) | -0.020(10) | -0.015(8) |
| HPE | 0.593(8) | 0.595(8) | 0.594(8) | 0.534(7) | **0.186**(5) | **0.094**(5) | HPE | 0.579(9) | 0.579(6) | 0.579(6) | 0.544(6) | 0.208(7) | **0.187**(5) |

- SEGEN: Model SEGEN proposed in this paper is based on the genetic algorithm and ensemble learning, which effectively combines the learned sub-network representation feature vectors from the unit models to generate the feature vectors of the whole network.

- LINE: The LINE model is a scalable network embedding model proposed in Tang et al. (2015), which optimizes an objective function that preserves both the local and global network structures. LINE uses a edge-sampling algorithm to addresses the limitation of the classical stochastic gradient descent.

- DEEPWALK: The DEEPWALK model Perozzi et al. (2014) extends the word2vec model Mikolov et al. (2013) to the network embedding scenario. DEEPWALK uses local information obtained from truncated random walks to learn latent representations.

- NODE2VEC: The NODE2VEC model Grover & Leskovec (2016) introduces a flexible notion of a node's network neighborhood and design a biased random walk procedure to sample the neighbors for node representation learning.

- HPE: The HPE model Chen et al. (2016) is originally proposed for learning user preference in recommendation problems, which can effectively project the information from heterogeneous networks to a low-dimensional space.

### 5.1.3 EVALUATION TASKS AND METRICS

The network representation learning results can hardly be evaluated directly, whose evaluations are usually based on certain application tasks. In this paper, we propose to use application tasks, network recovery and clustering, to evaluate the learned representation features from the comparison methods. Furthermore, the network recovery results are evaluated by metrics, like AUC and Precision@500. Meanwhile the clustering results are evaluated by Density and Silhouette. Without specific remarks, the default parameter setting for SEGEN in the experiments will be Parameter Setting 1 (PS1): sub-network size: 10, pool size: 200, batch size: 10, generation unit model number: 10, generation number: 30.

### 5.2 SOCIAL NETWORK DATASET EXPERIMENTAL RESULTS

The model training convergence analysis, and detailed analysis about the pool sampling and model learning parameters is available in the Appendix in Section 7.1. Besides these analysis results, we also provide the performance analysis of SEGEN and baseline methods in Tables 1-2, where the parameter settings are specified next to the method name. We provide the rank of method performance among all the methods, which are denoted by the numbers in blue font, and the top 5 results are in a bolded font. As shown in the Tables, we have the network recovery and community detection results on the left and right sections respectively. For the network recovery task, we change the ratio of negative links compared with positive links with values $\{1, 5, 10\}$, which are evaluated by the metrics AUC and Prec@500. For the community detection task, we change the number of clusters with values $\{5, 25, 50\}$, and the results are evaluated by the metrics Density and Silhouette.

Besides PS1 introduced at the beginning of Section 5.1, we have 4 other parameter settings selected based on the parameter analysis introduced before. PS2 for network recovery on Foursquare: sub-network size 50, pool size 600, batch size 5, generation size 50. PS3 for community detection on Foursquare: sub-network size 25, pool size 300, batch size 35, generation size 5. PS4 for network recovery on Twitter: sub-network size 50, pool size 700, batch size 10, generation size 5. PS5 for community detection on Twitter: sub-network size 45, pool size 500, batch size 50, generation size 5.

According to the results shown in Table 1, method SEGEN with PS2 can obtain very good performance for both the network recovery task and the community detection task. For instance, for the network recovery task, method SEGEN with PS2 achieves 0.909 AUC score, which ranks the second and only lose to SEGEN-HS with PS2; meanwhile, SEGEN with PS2 also achieves the second highest Prec@500 score (i.e., 0.872 for np-ratio = 1) and the third highest Prec@500 score (i.e., 0.642 and 0.530 for np-ratios 5 and 10) among the comparison methods. On the other hand, for the community detection task, SEGEN with PS3 can generally rank the second/third among the comparison methods for both density and

Table 3: Experiments on MNIST Dataset.

| Comparison Methods | Accuracy Rate% |
|---|---|
| SEGEN (CNN) | 99.37 |
| LeNet-5 | 99.05 Lecun et al. (1998) |
| gcForest | 99.26 Zhou & Feng (2017b) |
| Deep Belief Net | 98.75 Hinton et al. (2006) |
| Random Forest | 96.8 Zhou & Feng (2017b) |
| SVM (rbf) | 98.60 Decoste & Schölkopf (2002) |

Table 4: Experiments on Other Datasets.

| Comparison Methods | Accuracy Rate % on Datasets | | |
|---|---|---|---|
| | YEAST | ADULT | LETTER |
| SEGEN (MLP) | 63.70 | 87.05 | 96.90 |
| MLP | 62.05 | 85.03 | 96.70 |
| gcForest | 63.45 | 86.40 | 97.40 |
| Random Forest | 60.44 | 85.63 | 96.28 |
| SVM (rbf) | 40.76 | 76.41 | 97.06 |
| kNN (k=3) | 48.80 | 76.00 | 95.23 |

silhouette evaluation metrics. For instance, with the cluster number is 5, the density obtained by SEGEN ranks the second among the methods, which loses to SEGEN-LS only. Similar results can be observed for the Twitter network as shown in Figure 2.

By comparing SEGEN with SEGEN merely based on HS, BFS, DFS, NS, LS, we observe that the variants based on one certain type of sampling strategies can obtain relatively biased performance, i.e., good performance for the network recovery task but bad performance for the community detection task or the reverse. For instance, as shown in Figure 1, methods SEGEN with HS, BFS, DFS performs very good for the network recovery task, but its performance for the community detection ranks even after LINE, HPE and DEEPWALK. On the other hand, SEGEN with NS and LS is shown to perform well for the community detection task instead in Figure 1, those performance ranks around 7 for the network recovery task. For the Twitter network, similar biased results can be observed but the results are not identically the same. Model SEGEN combining these different sampling strategies together achieves relatively balanced and stable performance for different tasks. Compared with the baseline methods LINE, HPE, DEEPWALK and NODE2VEC, model SEGEN can obtain much better performance, which also demonstrate the effectiveness of SEGEN as an alternative approach for deep learning models on network representation learning.

### 5.3 EXPERIMENTS ON OTHER DATASETS AND UNIT MODELS

Besides the extended autoencoder model and the social network datasets, we have also tested the effectiveness of SEGEN on other datasets and with other unit models. In Table 3, we show the experimental results of SEGEN and other baseline methods on the MNIST hand-written image datasets. The dataset contains $60,000$ training instances and $10,000$ testing instances, where each instance is a $28 \times 28$ image with labels denoting their corresponding numbers. Convolutional Neural Network (CNN) is used as the unit model in SEGEN, which involves 2 convolutional layers, 2 max-pooling layers, and two fully connection layers (with a $0.2$ dropout rate). ReLU is used as the activation function in CNN, and we adopt Adam as the optimization algorithm. Here, the images are of a small size and no sampling is performed, while the learning results of the best unit model in the ending generation (based on a validation batch) will be outputted as the final results. In the experiments, SEGEN (CNN) is compared with several classic methods (e.g., LeNet-5, SVM, Random Forest, Deep Belief Net) and state-of-the-art method (gcForest). According to the results, SEGEN (CNN) can outperform the baseline methods with great advantages. The Accuracy rate obtained by SEGEN is $99.37\%$, which is much higher than the other comparison methods.

Meanwhile, in Table 4, we provide the learning results on three other benchmark datasets, including YEAST[1], ADULT[2] and LETTER[3]. These three datasets are in the traditional feature representations. Multi-Layer Perceptron (MLP) is used as the unit model in SEGEN for these three datasets. We cannot find one unified architecture of MLP, which works for all these three datasets. In the experiments, for the YEAST dataset, the MLP involves 1 input layer, 2 hidden layers and 1 output layers, whose neuron numbers are 8-64-16-10; for the ADULT, the MLP architecture contains the neurons 14-70-50-2; for the LETTER dataset, the used MLP has 3 hidden layers with neurons 16-64-48-32-26 at each layer respectively. The Adam optimization algorithm with $0.001$ learning rate is used to train the MLP model. For the ensemble strategy in these experiments, the best unit model is selected to generate the final prediction output. According to the results, compared with the baseline methods, SEGEN (MLP) can also perform very well with MLP on the raw feature representation datasets with great advantages, especially the YEAST and ADULT datasets. As to the LETTER dataset, SEGEN (MLP) only loses to gcForest, but can outperform the other methods consistently.

## 6 CONCLUSION

In this paper, we have introduced an alternative approach to deep learning models, namely SEGEN. Significantly different from the existing deep learning models, SEGEN builds a group of unit models generations by generations, instead of building one single model with extremely deep architectures. The choice of unit models covered in SEGEN can be either traditional machine learning models or the latest deep learning models with a "smaller" and "narrower" architecture. SEGEN has great advantages over deep learning models, since it requires much less training data, computational resources, parameter tuning efforts but provides more information about its learning and result integration process. The effectiveness of efficiency of SEGEN have been well demonstrated with the extensive experiments done on the real-world network structured datasets.

---

[1] https://archive.ics.uci.edu/ml/datasets/Yeast

[2] https://archive.ics.uci.edu/ml/datasets/adult

[3] https://archive.ics.uci.edu/ml/datasets/letter+recognition

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

# 7 APPENDIX

## 7.1 SOCIAL NETWORK DATASET EXPERIMENTAL ANALYSIS

In this part, we will provide experimental analysis about the convergence and parameters of SEGEN, including the sub-network size, the pool size, batch size and generation size respectively.

### 7.1.1 CONVERGENCE ANALYSIS

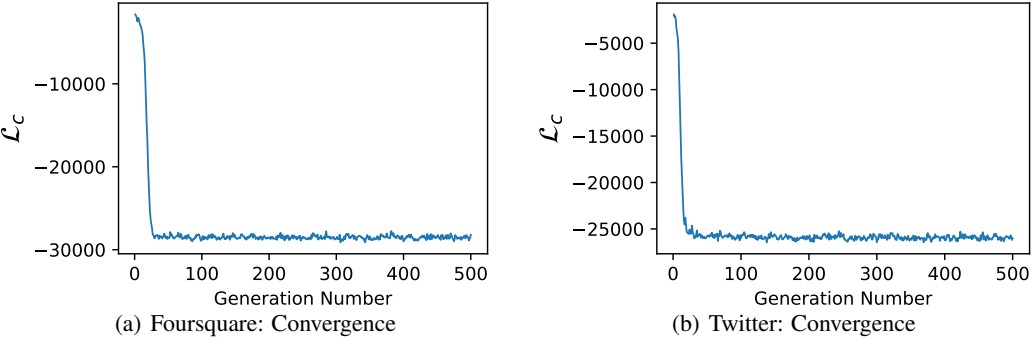

(a) Foursquare: Convergence         (b) Twitter: Convergence

Figure 3: Convergence Analysis on Foursquare and Twitter.

The learning process of SEGEN involves multiple generations. Before showing the experimental results, we will analyze how many generations will be required for achieving stable results. In Figure 3, we provide the introduced loss by the SEGEN on both Foursquare and Twitter networks, where the x axis denotes the generations and y axis represents the sum of introduced $\mathcal{L}_c$ loss on the validation set based on all these 5 different sampling strategies. According to the results, model SEGEN can converge within less 30 generations for the network representation learning on both Foursquare and Twitter, which will be used as the max-generation number throughout the following experiments.

### 7.1.2 POOL SAMPLING AND MODEL LEARNING PARAMETER ANALYSIS

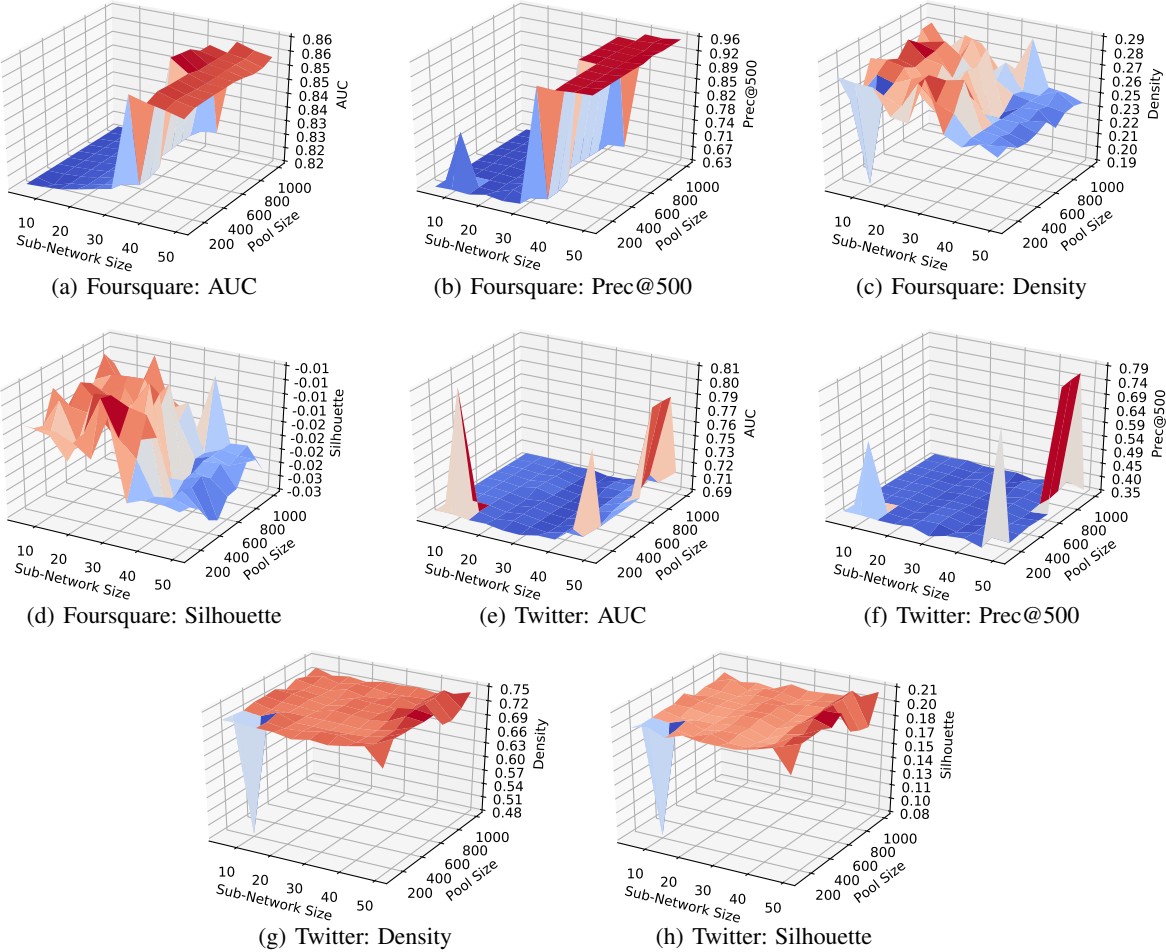

(a) Foursquare: AUC      (b) Foursquare: Prec@500      (c) Foursquare: Density

(d) Foursquare: Silhouette      (e) Twitter: AUC      (f) Twitter: Prec@500

(g) Twitter: Density      (h) Twitter: Silhouette

Figure 4: Sampling Parameter Analysis on Foursquare and Twitter.

In Figure 4, we show the sensitivity analysis about the network sampling parameters, i.e., sub-network size and the pool size, evaluated by AUC, Prec@500, Density and Silhouette respectively, where Figures 4(a)-4(d) are about the

Foursquare and Figures 4(e)-4(h) are about the Twitter network. The sub-network size parameter changes with values in $\{5, 10, 15, \cdots, 50\}$ and pool size changes with values in range $\{100, 200, \cdots, 1000\}$.

According to the plots, for the Foursquare network, larger sub-network size and larger pool size will lead to better performance in the network recovery task; meanwhile, smaller sub-network size will achiver better performance for the community detection task. For instance, SEGEN can achieve the best performance with sub-network size 50 and pool size 600 for the network recovery task; and SEGEN obtain the best performance with sub-network size 25 and pool size 300 for the community detection. For the Twitter network, the performance of SEGEN is relatively stable for the parameters analyzed, which has some fluctuations for certain parameter values. According to the results, the optimal sub-network and pool sizes parameter values for the network recovery task are 50 and 700 for the network recovery task; meanwhile, for the community detection task, the optimal parameter values are 45 and 500 respectively.

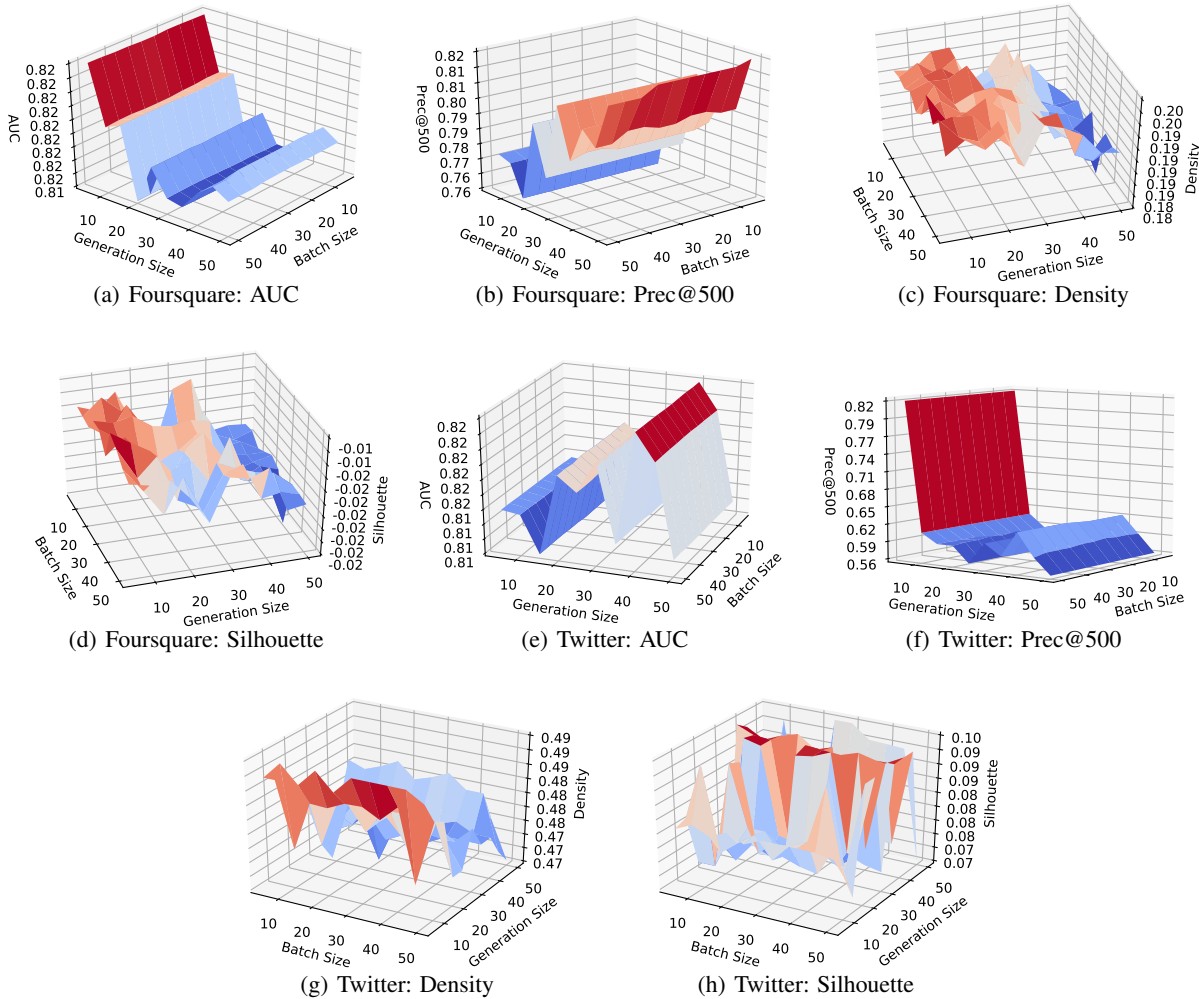

Figure 5: Batch and Generation Size Parameter Analysis on Foursquare and Twitter.

In Figure 5, we provide the parameter sensitivity analysis about the batch size and generation size (i.e., the number of unit models in each generation) on Foursquare and Twitter. We change the generation size and batch size both with values in $\{5, 10, 15, \cdots, 50\}$, and compute the AUC, Prec@500, Density and Silhouette scores obtained by SEGEN.

According Figures 5(a)-5(d), batch size has no significant impact on the performance of SEGEN, and the generation size may affect SEGEN greatly, especially for the Prec@500 metric (the AUC obtained by SEGEN changes within range [0.81, 0.82] with actually minor fluctuation in terms of the values). The selected optimal parameter values selected for network recovery are 50 and 5 for generation and bath sizes. Meanwhile, for the community detection, SEGEN performs the best with smaller generation and batch size, whose optimal values are 5 and 35 respectively. For the Twitter network, the impact of the batch size and generation size is different from that on Foursquare: smaller generation size lead to better performance for SEGEN evaluated by Prec@500. The fluctuation in terms of AUC is also minor in terms of the values, and the optimal values of the generation size and batch size parameters for the network recovery task are 5 and 10 respectively. For the community detection task on Twitter, we select generation size 5 and batch size 40 as the optimal value.

