# OpenReview forum: "SEGEN: SAMPLE-ENSEMBLE GENETIC EVOLUTIONARY NETWORK MODEL"
_ICLR.cc/2019/Conference_

### Official Review · AnonReviewer1 · 2018-11-04
**Interesting topic but several issues with the paper**

**Rating:** 4
**Confidence:** 4

**Review:**

This manuscript introduces SEGEN, a model based on Evolutionary Computation for building deep models. Interestingly, the authors define deep models in a different way. Instead of stacking several hidden layers one after the other (as in traditional deep learning models), SEGEN uses the idea of generations in evolutionary models (Genetic Algorithms or GA) and puts the unit models in the successive generations into layers, i.e., “evolutionary layer”. Each layer then performs the validation, selection, crossover, and mutation operations, as in GA. Another interesting point of the proposed method is that the choice of unit models in SEGEN can be traditional machine learning or recent deep learning models.
The paper touches an interesting topic and proposes a sound method. However, there are several issues with the paper. There are several ungrounded and untested claims, as well as many unclear points in the method.
-	In page 5, Section 4.2.4, the authors introduce the loss function used to define the fitness for the evolutionary model. It is not clear why they use the difference between the latent representations of the autoencoders (z) from pairwise nodes to define the loss. There are no motivations or discussion for this. Two different representations of two nodes may both be good (e.g., in terms of classification of data), but they do not have to be necessarily identical.
-	Given the loss defined in Section 4.2.4, it is not clear how the authors ran their model for MNIST and other datasets, for which they used CNN and MLP unit models. In CNN and MLP there is not latent representation z.
-	Based on the model descriptions in Section 4.2 (and its Subsections), the proposed method transfers the learned models in previous generations to the next ones. But there is no explanation if the new models are again fine-tuned on the data? For instance, take the autoencoders, for two different unit models, the cross-over operator defuses the variables (weights and bias) from the two selected models to create an offspring. There is no guarantee that the new autoencoder model works properly on the same dataset. As a naïve example, if there are correlated and redundant features in the data, different autoencoders may separately focus on one/some of these features. Defusing weights of the two autoencoders (built upon different aspects of the data) may most probably ruin the whole model.
-	There are four claims in the paper on the advantages of the proposed model, compared to other deep learning algorithms. None of these claims are discussed in depth or at least illustrated experimentally.
*** Less Data for Unit Model Learning. The authors could have reported the number of variables used in each model in the experiments. It is important to see with how many of a larger number of variables a traditional deep model can result in comparable results to SEGEN.
*** Less Computational Resources. The model operates in several generations and in each generation, many unit models are built. It is not fair to say and not clear how it can occupy less space or time complexity than a regular GCNN or MLP.
*** Less Parameter Tuning. Again experiments could clarify this issue.
*** Sound Theoretic Explanation. The authors only refer to (Rudolph 1994) for the performance bounds of their model and claim that since they are using GA they are better than other deep learning models. However, performance bounds for GA models are very shallow and proximal.
-	To calculate the computational complexity of the model, the authors analyzed the time for learning one unit model. However, in GA models, the complexity is calculated using the bounds on the number of times the fitness function is called since the fitness function is the most computationally intensive task (please see: Pelikan and Lobo 1999 ‘Parameterless Genetic Algorithm A Worst-case Time and Space Complexity Analysis’).
-	One of the main fallacies of GAs and evolutionary algorithms is that they may lead to premature convergence. This is very common, especially at the presence of trap functions, such as non-convex functions that real-world problems deal with (please see: Goldberg et al. 1991 ‘Massive Multimodality, Deception, and Genetic Algorithms’). There are no discussions/experiments on how SEGEN may overcome the premature convergence, or even if it converges at all.

---

> ### Author Response · Authors · 2018-11-14
> **Author Response to Reviewer 3**
>
> Thank you for your comments. Please find the response as follows. Hope we resolve your concerns and questions. Welcome to let us know if you have any other questions.
>
> <1> Section 4.2.4 in Page 5 on the loss function. We clarify that for each node we can compute its latent feature representation z with the auto-encoder model. However, graph embedding is slightly different form other existing embedding problems, since the nodes are connected. Generally, in graph embedding, we may hope the learned representation features can capture the network structure: connected nodes will have closer representations. Therefore, given two nodes, v_i, v_j, if they are connected, i.e., s_ij = 1, then we may want to project them into close regions; if they are not connected, i.e., s_ij = 0, then we will not count the loss introduced by them, i.e., projecting them to any regions will not matter any more.
>
> <2> The auto-encoder model as well as the z vectors is for the network embedding task only (auto-encoder as the base model, we further consider the graph connections). We use it as an example to introduce the overall SEGEN framework settings. The task and the unit models used in it can be changed to any other models, where the detailed loss function and the descriptions will be different. When it comes to CNN+MNIST or MLP+OtherDatasets, we will learn CNN unit models and MLP unit models instead, which will not contain the z vectors or the loss function in section 4.2.4. Instead, we will have some other loss functions on the CNN output, e.g., the cross-entropy on the CNN outputs compared against the true labels.
>
> <3> New models are fine-tuned? We claim that the new models will be fine-tuned in the next generation on the dataset. Since in the new generation, the first step is to learn the unit models before involving them in the genetic evolutionary part. They will be trained on the training set.
>
> <4> We clarify that we have provide the proof ready, but due to the limited space we remove many important proofs. We demonstrate that via GA and ensemble, we can achieve better performance. The reviewer is suggested to refer to Section 4 in the recent article (https://arxiv.org/abs/1805.07500) for more information. Especially Equation 14 in that article, it indicates that via generations the learning loss will be non-increasing.
>
> <5> We clarify that our time and space cost analysis is not for one unit model, it is for the whole SEGEN model with multiple generations. The time complexiety provided before Section 4.4.3 contains K as the generation number, m as the population size.
>
> <6> We clarify that for the SEGEN model introduced in this part, the fitness function computation is not the most computationally intensive task actually, since the unit model learning with gradient descent in Section 4.2.3 will be much time consuming. The time costs in learning the models may grow exponentially as the model size (I mean the input data size) increases. That is the reason we try to sample the sub-graphs in this paper instead to lower down the time cost compared against the existing deep models (it is also our main contribution and advantages). Fitness function computation is the most computationally intensive task in (Pelikan and Lobo 1999 mentioned in your comments), mainly compared against the mutation and cross operations. Here, the learning settings change to the deep learning model learning + evolutionary. Compared against model learning, GA based evolutionary time cost is not significant any more, not to mention the fitness function computation part.
>
> <7> As to the convergence part, we clarify that we have provide the proof ready, but due to the limited space we remove many important proofs. The reviewer is also suggested to refer to Section 4 in the recent article (https://arxiv.org/abs/1805.07500) for more information. Especially Equation 14 in that article, it indicates that via generations the learning loss will be non-increasing, and it will converge generations by generations.

---

> > ### Comment · AnonReviewer1 · 2018-11-21
> > **Response to the authors' rebuttal**
> >
> > Thanks for the response. Some of the concerns are resolved through the rebuttal. Here are some important issues:
> > - Point <2> needs to be clarified in the manuscript.
> > - Point <3> is not raised or mentioned in the manuscript! Needs clear clarification, and then ablation studies to show how they help.
> > - With respect to point <4>, the limited space notations cannot be a good reason. The authors have the option of providing supplementary material, which does not have any space limitations or constraints, and the authors did not use that. Furthermore, referring to external material (other papers, or papers on arxiv) in the rebuttal to answer a direct concern of the reviewers does not seem correct. If something is required for the paper to be understood and to justify the correctness of the work, it should be provided in the paper or the supplementary material!
> > - For point <6>, the authors seem not to understand what the fitness function means here. Fitness in the GA (in your setting) would be defined as evaluating a single unit model. The evaluation consists of building a unit model, fine-tuning, and evaluating. All these are functions of the data size (or portions of it). Then, the deep generation by generation (k \times m) adds to it and is a constant factor.
> > - For point <7>, the authors again refer to an external publication, without enough discussions and justifications here.
> >
> > If every question raised by this reviewer (and also other reviewers) can be answered by other papers that the authors keep referring to in the rebuttal, then, what is the need for publishing this new paper!?
> >
> > Although this reviewer appreciates the response from the authors, I still think the paper is not mature and is not ready for publication.

---

### Official Review · AnonReviewer2 · 2018-11-04
**Using Subsampling + Genetic Algorithm for Network Embedding**

**Rating:** 5
**Confidence:** 5

**Review:**

This paper proposes to subsample a large network into sub-networks, learn a network model (autoencoder) from each subgraph, perform crossover and mutation operations over the network parameters of different model pairs, and combine the latent representations following the ensemble idea.

The paper is clearly presented. Originality and significance is limited. Putting the three knowns components - subsampling, generation algorithm and ensembling together seems to be the main contribution of this paper. However, the ways of doing subsampling, performing the crossover and mutation operations and doing the ensembling are relatively straightforward ways of applying them. The fact that combining them to obtain better results is not a surprising result. And according to the experimental results, it is not clear how the gain in performance is resulted and to what extent each of the three components is contributing. For instance, I just guess the combination of subsampling + existing network embedding methods (LINE/DeepWalk/...) + ensembling may also give good results. Currently, the performance comparison is done with the original forms of LINE and DeepWalk. That makes the empirical results not very convincing to explain the key strengths of this work.

+ve:
`1. The paper is clearly presented.
2. The design is reasonable one.
3. A number of benchmark datasets are used for the evaluation.

-ve:
1. The originality and significance is limited.
2. The performance comparison should be done with references to more competitive candidates as explained above.
3. The nodes of different sub-networks are essentially projected to different embedding spaces. The validity and interpretation of performing the crossover operation on two different models (two different embedding spaces) will need more justifications.
4. The proposed methodology is not an end-to-end. The ensembling being evaluated is just simple addition.
5. The paper claims that "The unit learning model, genetic algorithm and ensemble learning can all provide the theoretic foundation for SEGEN, which will lead to sound theoretic explanation of both the learning result and the SEGEN model itself". Individually being sound does not imply that the way to combine them is sound. Currently, I cannot see the uniqueness of this particular combination.

---

> ### Author Response · Authors · 2018-11-14
> **Author Response to Reviewer 2**
>
>
> Thank you for your comments. Please find the response as follows. Hope we resolve your concerns and questions.
>
> <1> First of all, we need to re-clarify the contributions of this article.
>
> <1.1> For the learning settings with extremely large-sized but small-numbered data instances (i.e., each data instance is large, but the total number of available data instance is small), training large and deep neural networks is an impossible mission. In this paper, we propose a solution to such a problem.
>
> <1.2> This paper doesn’t really like existing deep learning works focusing in stacking components together. It is not a good idea to interpret our contribution as “putting three known components together”. According to <1.1>, to solve the lack of data instance problem, we propose to divide the large graph into small-sized sub-graphs. To ensure the sub-graphs can capture the properties of the large graph, we use different sampling methods. Meanwhile, in the training process, to ensure the learning effectiveness, we also introduce a new learning framework, with both the gradient descent based algorithm with the genetic algorithm. As to the ensemble part, it not merely because we decompose the large graph into smaller graphs. The main reason is we have a group of small models, each one is trained on sub-graphs achieved by a sampling method, we need to integrate the outputs of these models together.
>
> <1.3> The model learning part of the model proposed in this paper is based on both the gradient decent based algorithms (for each unit model), as well as the genetic algorithm (between different generations). This part should be notable to the reader and the reviewer.
>
> <2> How the gain in performance is resulted? We clarify that with a small number of large data instance inputs, we cannot train effective deep models due to the lack of data instances. By decomposing the large graphs into smaller ones, we will be able to learn effective model variables.
>
> <3> sampling+existing embedding model+ensemble should also be useful. The answer is yes, since our framework and our new learning algorithm is useful, replacing the auto-encoder based embedding algorithm with the other shallow or deep embedding models should also work fine. The reviewer is suggested to read the paper again to understand what we do, so as to understanding our contributions, instead of treating it is a combination of sampling+ensembling.
>
> <4> Originality and significance is limited: This is the first paper to introduce the genetic evolutional neural network! Different from the existing deep learning model works, we propose a novel network model trainable with a small-set of extremely large data instances. We introduce a new model learning algorithm with both gradient decent based algorithms and the genetic algorithm. I assume the reviewer cannot find another paper with these two novelty and contributions.
>
> <5> The baseline methods, LINE, DeepWalk, Node2Vec, and HPE are the state-of-the-art methods in network embedding introduced in recent years.
>
> <6> Crossover on models. Crossover operation is to help learn a much better unit model actually. Based on the gradient descent algorithms, we will be able to learn good unit models. However, once the unit model achieving the local optima, it cannot be further improved any more. Genetic algorithm (including crossover, mutation, etc.) allows the models to jump out from the local optima and achieve better performance. The generated child models will be updated with gradient decent algorithm again to achieve the local optimas. Will this make the models worse? The answer is it is possible. However, in the proposed architecture, we will select top m models among the parent models and the newly generated child models. If the child models are bad, they will not be selected for the next generation. In other words, we can ensure the crossover will not degrade the learning performance of the unit models for the next generation. The readers and reviewers are suggested to refer to the recent article (https://arxiv.org/abs/1805.07500) to understand the advantages of incorporating genetic algorithm into the model optimization part.
>
> <7> The method is not end-to-end. Since the genetic algorithm involves crossover and mutation, this part involves probabilities into the model, it is impossible to train the crossover and mutation operations with the existing error-backpropagation algorithm. In other words, training the method in an end-to-end is an impossible mission.
>
> <8> GA + Ensemble together not sound. We clarify that we have provide the proof ready, but due to the limited space we remove many important proofs. We demonstrate that via GA and ensemble, we can achieve better performance. The reviewer is suggested to refer to Section 4 in the recent article (https://arxiv.org/abs/1805.07500) for more information. Especially Equation 14 in that article, it indicates that via generations the learning loss will be non-increasing.

---

> > ### Comment · AnonReviewer2 · 2018-11-25
> > **response to rebuttal**
> >
> > I appreciate the authors' effort in clarifying their contributions and the explanation.
> >
> > <1.1> The solution is subsampling a large graph and the ways of subsampling have been adopted by others. As mentioned in the comments, noveling is limited.
> > <1.2> Maybe the use of the word "component" confuses the authors. The three "components" refer to subsampling, GA and ensemble learning.
> > <3> After reading your response, I still think that the key contribution of the paper is combination of subsampling, GA and simple ensemble learning.
> > <4> Using GA for global optimization has long been proposed. The use of GA for evolving neural networks has also been considered and tested with different success. And the way being adopted by the authors is not particularly novel.
> > <5> As explained, combinations of subsampling + existing network embedding methods (LINE/DeepWalk/...) + ensembling may also give good results.
> > <6> As explained, the fact that GA was proposed for global optimization is a well known fact.
> > <7> I think this is an open-ended question leaving for the authors to think about.
> > <8> "GA + Ensemble together not sound" Again I think the authors misunderstand some of my comments and hard to provide further comments here.
> >
> > After reading the rebuttal, I think I still uphold my original rating.

---

### Official Review · AnonReviewer3 · 2018-11-04
**Evolutionary part is not clear**

**Rating:** 5
**Confidence:** 2

**Review:**

The paper introduces Sample-Ensemble Genetic Evolutionary Network, which adopts a genetic-evolutionary learning strategy to build a group of unit models. Explanation on the evolutionary network part is not enough. For example, there is no clear explanation on how chromosomes are defined. Also, detailed analysis on computational aspect is needed.

---

> ### Author Response · Authors · 2018-11-14
> **Author Response to Reviewer 1**
>
> Thank you for your comments. Please find the response as follows. Hope we resolve your concerns and questions. Welcome to let us know if you have any other questions.
>
> <1> We clarify that we introduce the model chromosomes as the variables of the models. You can refer to the last two sentences in section 4.2.1. as well as section 4.2.5. We also paste the sentences as follows.
> 4.2.1: Formally, the variables involved in each unit model, e.g., M_i^1, can be denoted as vector θ_i^1, which covers the weight and bias terms in the model (which will be treated as the model genes in the evolution to be introduced later) ).
> 4.2.5: For the k_th pair of parent unit model (M_i^1,M_j^1)k ∈ P^1, we can denote their genes as their variables θ_i^1,θ_j^1 respectively (since the differences among the unit models mainly lie in their variables), which are actually their chromosomes for crossover and mutation.
>
> <2> We clarify that we introduce the computational analysis in Section 4.4, including its performance analysis, space and time cost analysis, as well as advantages analysis.

---

### Meta-Review · Area_Chair1 · 2018-12-17
**Interesting topic but requires more work**

**Confidence:** 5
**Recommendation:** Reject

**Metareview:**

This paper endeavors to combine genetic evolutionary algorithms with subsampling techniques. As noted by reviewers, this is an interesting topic and the paper is intriguing, but more work is required to make it convincing (fairer baselines, more detailed / clearer presentation, ablation studies to justify the claims made int he paper). Authors are encouraged to strengthen the paper by following reviewers' suggestions.